# Preserving a qubit during state-destroying operations on an adjacent qubit at a few micrometers distance

Sainath Motlakunta [1,2] ✉, Nikhil Kotibhaskar[1,2], Chung-You Shih [1,2], Anthony Vogliano[1,2], Darian McLaren[1,2], Lewis Hahn[1,2], Jingwen Zhu[1,2], Roland Hablützel[1,2] & Rajibul Islam [1,2]

Protecting qubits from accidental measurements is essential for controlled quantum operations, especially during state-destroying measurements or resets on adjacent qubits, in protocols like quantum error correction. Current methods to preserve atomic qubits against such disturbances waste coherence time, extra qubits, and introduce additional errors. We demonstrate the feasibility of in-situ state-reset and state-measurement of trapped ions, achieving >99.9% fidelity in preserving an 'asset' ion-qubit while a neighboring 'process' qubit is reset, and >99.6% preservation fidelity while applying a detection beam for 11 µs on the same neighbor at a distance of 6 µm. This is achieved through precise wavefront control of addressing optical beams and using a single ion as both a quantum sensor for optical aberrations and an intensity probe with >50 dB dynamic range. Our demonstrations advance quantum processors, enhancing speed and capabilities for tasks like quantum simulations of dissipation and measurement-driven phases, and implementing error correction.

Programmable many-body quantum systems are an excellent platform for quantum information processing (QIP), including simulation of complex quantum phenomena and quantum computing. Full programmability requires both coherent and incoherent control, such as state resets (initialization) and state measurements at the level of its individual building blocks[1,2]. Coherent dynamics are, in principle, reversible, while incoherent operations generally constitute irreversible quantum measurements. The ability to perform measurements and resets on a subsystem in the middle of coherent dynamics ('mid-circuit measurements and resets') is a powerful tool for simulating new classes of quantum phenomena such as measurement-driven quantum phase transitions[3–9] and executing quantum error correction protocols[1,10,11]. A primary challenge[12] of subsystem mid-circuit measurement and reset is the accidental quantum measurement (AQM) of the remaining system during the process, leading to irreparable decohering errors. Specifically, in the context of atomic quantum systems like trapped ions, the 'process' qubit undergoes a reset or measurement through resonant laser beam illumination (Fig. 1a). The neighboring 'asset' qubit may absorb light, either scattered from this process qubit (inter-ion scattering) or leaked from the laser beam (intensity crosstalk), leading to a finite probability of AQM ($P_{AQM}$). This probability can be prohibitively high, as the typical inter-atomic separation is comparable to the optical resolution. To mitigate the high probability of AQM in atomic QIP experiments, various strategies are employed. These include physically separating atoms through shuttling[13–16], utilizing different atomic species[17–20], implementing delayed measurement schemes by using additional qubits of the same species[3], hiding qubits in states outside the computational Hilbert space[21–25], and employing other suppression techniques[12]. However, these techniques waste valuable resources, such as circuit time and extra qubits, and also introduce new errors, such as those resulting from motional heating or imperfect coherent operations.

[1]Institute for Quantum Computing, University of Waterloo, Waterloo, ON N2L 3G1, Canada. [2]Department of Physics and Astronomy, University of Waterloo, Waterloo, ON N2L 3G1, Canada. ✉e-mail: smotlaku@uwaterloo.ca

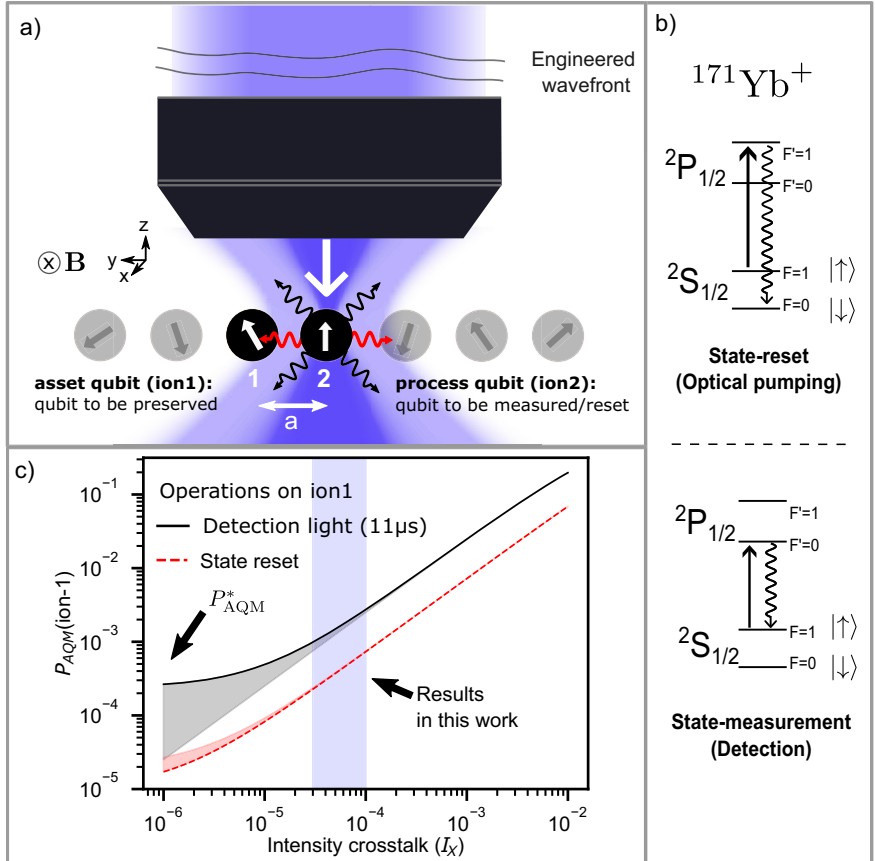

**Fig. 1 | Accidental quantum measurement (AQM) of neighboring qubits. a** While addressing a 'process' qubit(ion2) in a trapped ion chain, an 'asset' qubit (ion1) at a distance $a$ away may be accidentally measured by photons that are either scattered from ion2 (red wavy lines) or from intensity crosstalk due to imperfect optical addressing (lightly shaded violet). AQM from imperfect optical addressing can be minimized by engineering the wavefronts incident on the microscope objective. **b** Atomic transitions in $^{171}Yb^+$ (Zeeman splitting not shown) for relevant incoherent processes. The ground state hyperfine levels $S_{1/2}|F=0, m_F=0\rangle$ and $S_{1/2}|F=1, m_F=0\rangle$ are assigned as the $|\downarrow\rangle$ and $|\uparrow\rangle$ of the effective spin-1/2 object or a qubit, respectively. Top - a quantum state is reset through optical pumping into $|\downarrow\rangle$. Bottom - a quantum state is measured in $\{|\downarrow\rangle, |\uparrow\rangle\}$ basis by detecting state-dependent fluorescence[29] from the cycling transition. **c** Calculated probability of

AQM ($P_{AQM}$) of the asset qubit (ion1) as a function of intensity crosstalk ($I_X$). Here, $P_{AQM}$ is estimated from the asset qubit's infidelity after a state detection or reset on the process qubit. The fidelity[28] is estimated with respect to $|\uparrow\rangle$ to represent the worst-case scenario (Supplementary Note 3). For this figure, we choose $a = 6\,\mu m$, and $I_2 = I_{sat}$ (the saturation intensity of the transition). For low crosstalk regime ($I_X < 1 \times 10^{-5}$), inter-ion scattering sets a fundamental limit, $P_{AQM}^*$, which can vary (shaded region) depending on the geometric properties of the system, such as the orientation of the magnetic field ($\vec{B}$) defining the quantization axis(see Methods). The results presented in this manuscript are in the regime with $I_X \lesssim 8 \times 10^{-5}$, leading to $P_{AQM} < 4 \times 10^{-3}$ for state reset, and $P_{AQM} < 1 \times 10^{-3}$ with a detection beam applied for $11\,\mu s$[13].

Here, we demonstrate that in-situ reset and measurement of trapped ions is feasible. We achieve $P_{AQM} < 1 \times 10^{-3}$ of the asset qubit while resetting (Fig. 1b) the process qubit placed at a distance of 6 μm, and $P_{AQM} < 4 \times 10^{-3}$ while applying detection light on the process qubit for experimentally demonstrated[13] fast detection times of 11 μs. These low probabilities of AQM correspond to the preservation of the quantum state of the asset qubit with fidelities exceeding 99.9% and 99.6% for the reset and measurement processes, respectively. Here, we ignore any additional measurement arising due to the entanglement of qubits in the system. Our measured low $P_{AQM}$ is enabled by low relative intensity crosstalk of $I_X < 1 \times 10^{-4}$ that is maintained across a large spatial region of $> 400\,\mu m$, suitable to address $> 50$ ions. Here, the intensity crosstalk $I_X = I_1/I_2$ is defined as the ratio of the intensity of the probe beam on the asset qubit ($I_1$) to that on the process qubit ($I_2$).

Inter-ion scattering sets a fundamental limit to $P_{AQM}$ of $P_{AQM}^* \propto 1/a^2$ in the regime where inter-ion spacing $a$ is much larger than the wavelength of radiation. In addition, the exact value of $P_{AQM}^*$ will depend on the magnetic field (quantization axis) configuration (see Methods). We demonstrate through theoretical estimations and experimental measurements that the $P_{AQM}$ with our measured

intensity crosstalk at the asset qubit approaches but is not yet limited by $P_{AQM}^*$ (Fig. 1c).

We employ a holographic system for precise control of the laser beam's wavefront, correcting aberrations in the optical system. Our relative intensity crosstalk at the ion is a hundred times lower than what was attained in previous efforts using this technology[26]. This reduction in crosstalk is crucial for high-fidelity, in-situ mid-circuit measurements and resets, and arises from two key advancements in our methods. Firstly, we utilize the qubit state's high sensitivity to a reset beam for intensity measurement, offering enhanced precision and spatial resolution for aberration characterization. Secondly, we assess intensity crosstalk using the wide dynamic range of the coherence time $T_2^*$, obtained from Ramsey interferometry on the qubit. The coherence time $T_2^*$ varies greatly with beam intensity, turning the qubit into a highly efficient intensity sensor with a dynamic range exceeding 50 dB. This approach verifies the extremely low crosstalk.

Our method, based on robust optical engineering instead of unique trapping structures, can be applied to different atomic QIP systems. The high-fidelity outcomes we have achieved, pave the way for investigating new protocols in measurement-driven quantum simulation[4] and open quantum systems. These include quantum

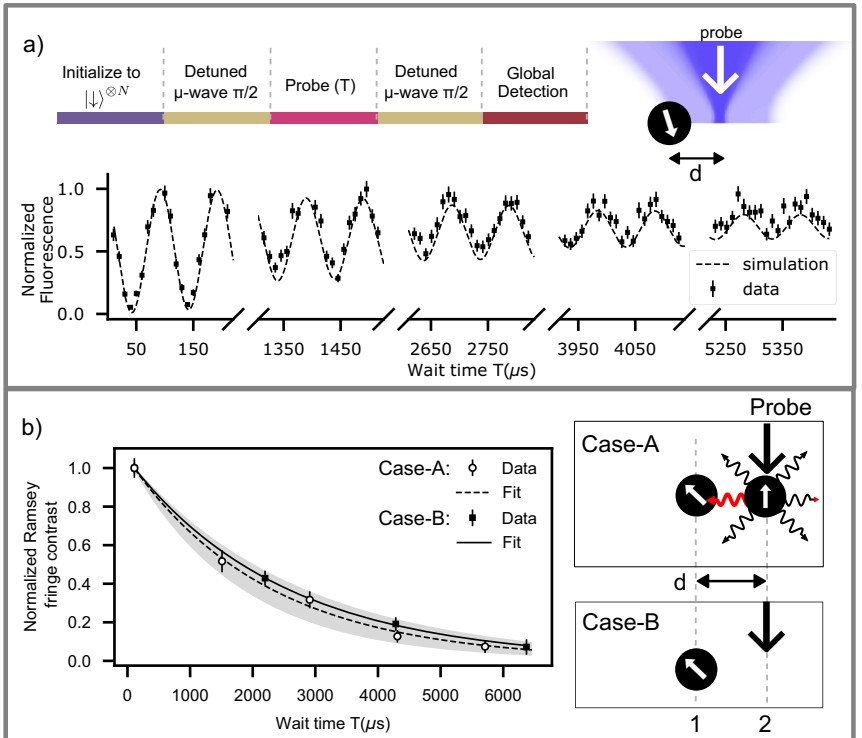

**Fig. 2 | AQM characterization scheme. a** (Top) Ramsey interferometric protocol (Supplementary Note 2) to measure qubit coherence time $T_2^*$ when the incoherent probe beam is applied for time $T$ at a distance $d$. (Bottom) Ramsey fringes in normalized fluorescence originating from the detuning between the microwave source and the qubit frequency. $T_2^*$ is extracted from the decay in Ramsey fringe contrast. Here, the background-subtracted fluorescence counts are measured during the global detection step and are normalized with respect to the counts from $|\uparrow\rangle$. The data shown here are for a single ion ($N = 1$) illuminated with the state-detection probe light ($I = 1.25(16)I_{sat}$, waist $w = 1.50(5)$ μm) at a distance of $d = 6.0(3)$ $w = 9.0(4)$ μm and Ramsey detuning of 10 kHz. Error bars indicate standard error from 200 experimental repetitions. Intensity crosstalk $I_X$ is estimated from numerical simulations of the master equation from the measured $T_2^*$. We find, using numerical simulations solving the master equation of the system (dashed line,

Supplementary Note 5), that the intensity cross-talk, $I_X = 3.4(6) \times 10^{-5}$ for this data. **b** Comparison of Ramsey fringe decay profiles between case-A: an ion located at the probe beam focus ($N = 2$), and case-B: no ion at probe beam focus ($N = 1$). Data points represent Ramsey fringe contrast measured over two fringes, and the fits are exponential decay with $T_2^*$ as a fitting parameter. The Ramsey fringe contrast is normalized with the contrast measured at $T = 0$. Error bars denote standard deviation in estimating Ramsey fringe contrast, using 20 bootstrapping repetitions from 200 measurements (Supplementary Note 7). The shaded region indicates fluctuations of experimental settings over periodic calibration of the probe beam location with respect to ion2 (Supplementary Note 4E) for case-A. $T_2^*$ values measured for case-A and case-B lie within the error bounds, indicating that the decoherence is limited by the intensity crosstalk and not by inter-ion scattering.

simulations with localized dissipation and measurements, as well as quantum reservoir engineering. In addition, in-situ site-selective state reset will facilitate sympathetic cooling of a subsystem without requiring multiple atomic species[27]. This effectively enhances both the scalability and extends the capability to perform longer QIP protocols. The in-situ operations in our approach make for scalable, simple, robust and fast QIP protocols, offering advantages over other error mitigation strategies like ion shuttling and using additional qubits for mid-circuit measurements.

## Results

We use the ground state hyperfine levels of $^{171}$Yb$^+$ ions trapped and Doppler-cooled in a 'four-rod' Paul trap as $|\downarrow\rangle$ and $|\uparrow\rangle$ of the effective spin-1/2 object or a qubit (Supplementary Note 1). These ions are individually probed through an addressing system with an effective numerical aperture(NA) of 0.16(1). The optical aberrations in the system are characterized (see Methods) using a single ion as a quantum sensor. Using a measured aberration phase profile, a Fourier hologram employed on a digital micromirror device (DMD) is programmed to create a diffraction-limited Gaussian beam of waist $w = 1.50(5)$ μm in the ion plane. This beam is positioned at a programmable distance $d$ from the ion while minimizing intensity leakage onto neighboring ions.

In the regime where the probability of the asset qubit accidentally scattering a photon, $P_{AQM} \ll 1$, we find numerically that

the infidelity of the asset qubit is a good estimate of $P_{AQM}$ (Fig. 1c). The fidelity[28] of preserving the state of the asset qubit is estimated from fringe contrast in a Ramsey interferometry experiment (Fig. 2a.) We measure the fringe decay (decoherence) time $T_2^*$ of the asset qubit(ion1) and estimate (Supplementary Note 3) the fidelity of preserving its state after measurement or reset on the process qubit (ion2) from,

$$F_{1|2} = \frac{2}{3}\exp\left[-\frac{\tau(\text{ion2})}{T_2^*(\text{ion1})}\right] + \frac{1}{3}. \quad (1)$$

Here, $\tau(\text{ion2})$ is the time for which the resonant probe beam illuminates the process qubit. From the measured $T_2^*$, we estimate the intensity of probe light sampled by the asset qubit. The long quantum memory of the asset qubit (without any probe light $T_2^* \gg 200$ ms is much longer than the results in Fig. 2b) enables it to act as a sensitive, high-dynamic range sensor for intensity crosstalk.

To distinguish the decoherence caused by inter-ion scattering and the imperfect optical addressing, we perform the above Ramsey measurements for two different cases. case-A uses two ions, separated by a distance $d$, with a probe beam addressing ion2. case-B uses only one ion with a probe beam located (Supplementary Note 4E) at the same distance $d$ from the ion (Fig. 2b). For $d = 6w$ (9 μm), we find that the Ramsey fringe decay time ($T_2^*$) for both experiments is

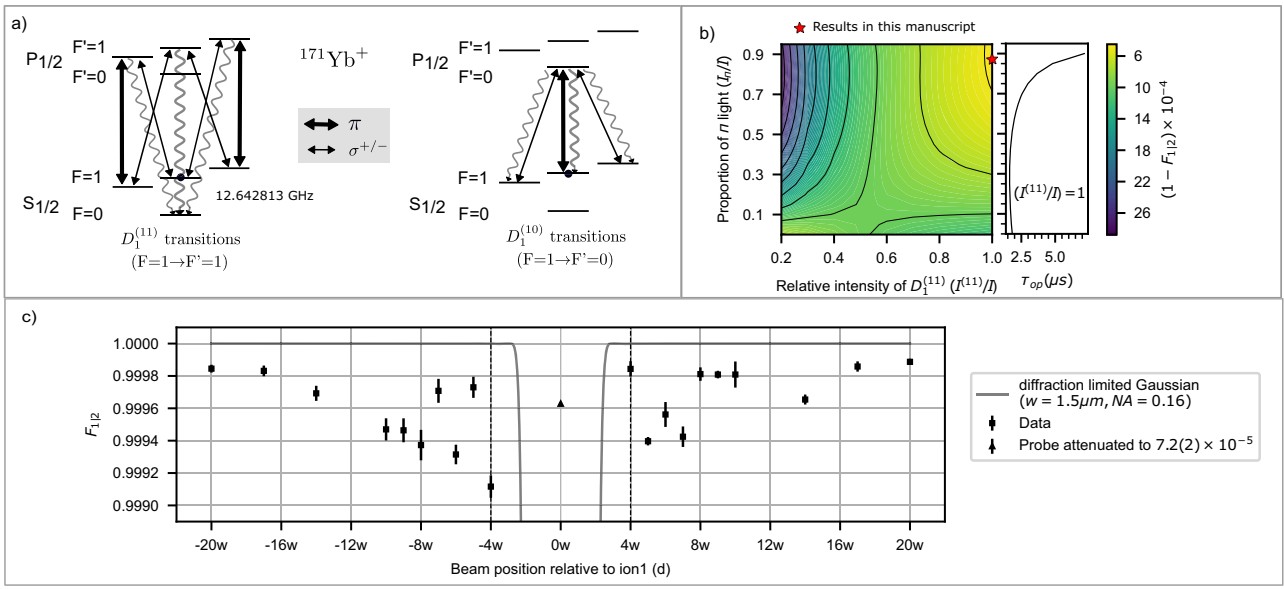

**Fig. 3 | Fidelity ($F_{1|2}$) of preserving ion1 for state reset light at ion2 location.** **a** Excitation and decay mechanisms for the $D_1^{(11)}$ and $D_1^{(10)}$ transitions in $^{171}$Yb$^+$ ion initialized in state $|\uparrow\rangle$[29], for various polarizations (thick arrows representing $\pi$ and thin arrows representing $\sigma^\pm$). The $D_1^{(11)}$ transitions contribute to state reset via optical pumping, although any residual $D_1^{(10)}$ light (e.g., from frequency modulation via an electro-optic modulator[29]) may degrade $F_{1|2}$. **b** Calculated $F_{1|2}$ for the state reset process as a function of the ratio of the intensity of $D_1^{(11)}$ component $I^{(11)}$ to the total intensity $I$ (where $I = I^{(11)} + I^{(10)}$ with $I^{(10)}$ indicating $D_1^{(10)}$ component) and ratio of the intensity of $\pi$ polarization $I_\pi$ to the total intensity $I$ (where $I = I_\pi + I_{\sigma^+} + I_{\sigma^-}$ with equal intensities in $\sigma^+$ and $\sigma^-$ polarizations) Here, $F_{1|2}$ is calculated using numerical simulations of the master equation (Supplementary Note 5) under the conditions of $I_2 = 1.25 I_{\text{sat}}$ and $I_X = 5 \times 10^{-5}$. The red star marker indicates the parameters used to measure $F_{1|2}$ in (c). Additionally, the plot on the right (sharing the same vertical

axes) shows an estimation of state reset times $\tau_{\text{op}}$(ion2) as a function of $I_\pi/I$ for $I^{(11)}/I = 1$. **c** $F_{1|2}$ vs $d$ expressed in multiples of the beam waist $w$ (case-B in Fig. 2b). Here, $w = 1.50(5)$ μm is the Gaussian beam waist for the addressing beam. Error bars denote standard deviation in estimating $F_{1|2}$, using 20 bootstrapping repetitions from 200 measurements (Supplementary Note 7). The estimated infidelity ($1$-$F_{1|2}$) due to the inherent decoherence of the qubit in the absence of the probe light is $< 3 \times 10^{-5}$. For calibrating crosstalk $I_X$, we measure $F_{1|2}$ for a probe beam with relative intensity attenuated to $7.2(2) \times 10^{-5}$ addressing ion1 (triangle marker at $d = 0$). For comparison, $F_{1|2}$ is calculated (solid gray line) for a diffraction-limited (Numerical Aperture NA $= 0.16$) Gaussian beam of beam waist $w = 1.50$ μm. $F_{1|2}$ is $> 99.9\%$ for $d \geq 4w$ (see discussion). For these measurements, $I_2 = 1.25(16) I_{\text{sat}}$, $I_\pi/I = 0.86$, $I^{(11)}/I = 1$, $\tau_{\text{op}} = 9.73(7)$ μs.

indistinguishable (within the experimental fluctuations)(Fig. 2b). This verifies that the inter-ion scattering is not the major source of deco- herence in our experiment. Thus we could use a single-ion (case-B) to quantify the fidelity $F_{1|2}$ in our addressing scheme, which greatly sim- plifies the measurement scheme.

The process duration $\tau$ and the decoherence time $T_2^*$ in Eq. (1) may have different dependence or optima over optical parameters (such as polarizations, spectral purity, etc.). In the following experiments, we maximize $F_{1|2}$ by maximizing the fraction of light contributing to the process (state reset or measurement) while minimizing (where possi- ble) the fraction of light that accidentally measures the asset qubit.

### Site-selective state reset
The process of state reset through optical pumping is done by using a probe that drives the transition $D_1^{(11)}$ from $S_{1/2}|F=1\rangle$ to $P_{1/2}|F'=1\rangle$[29] (Fig. 3a). We choose the process time $\tau_{\text{op}}$(ion2) $= 7T_1$, where $T_1$ indi- cates the time at which the normalized ion fluorescence of ion2 drops to $1/e$ compared to its initial value. This will ideally reset the quantum state of the process qubit to $|\downarrow\rangle$ with the fidelity of $1 - e^{-7} = 0.999$. Note that spectral components of light apart from $D_1^{(11)}$ may decohere the asset qubit while not contributing to the reset on the process qubit (ion2). For example, optical pumping light derived by frequency modulation (electro-optic modulation) employed in typical ion trap experiments[29] contains residual $D_1^{(10)}$ component (the spectral com- ponent used for detection). This $D_1^{(10)}$ component will increase $P_{\text{AQM}}$ and hence reduce $F_{1|2}$, as shown by numerical simulation data in Fig. 3b where $F_{1|2}$ is maximized for a probe with relatively higher intensity in $D_1^{(11)}$ component ($I^{(11)}$).

Probe light of different polarizations has unequal contributions to the asset qubit's fidelity $F_{1|2}$ for the case of state reset. For $D_1^{(11)}$

transition, since the $|\uparrow\rangle = S_{1/2}|F=1, m_F = 0\rangle$ to $P_{1/2}|F'=1, m'=0\rangle$ is dipole forbidden, the component with $\pi$ polarization with intensity $I_\pi$ doesn't contribute to the AQM of the asset qubit. So, for the case of probe with a total intensity $I$, where $I^{(11)}/I = 1$, $F_{1|2}$ increases with $I_\pi/I$ (Fig. 3b). This increase in $F_{1|2}$ comes at the cost of increasing the state reset $\tau_{\text{op}}$ of ion2. For the case of $I^{(11)}/I \neq 1$, the $\pi$ polarizations of the $D_1^{(10)}$ transition component of the probe still contribute to the AQM of the asset qubit. Hence an increase in $I_\pi/I$ decreases $F_{1|2}$ as the light with $\pi$ polarization only contributes to AQMs of the asset qubit but not to the reset of the process qubit(Fig. 3b).

With the polarization of the state reset beam optimized, we characterize $F_{1|2}$ (in case-B configuration) as a function of beam posi- tion relative to the ion ($d$) (Fig. 3c) and observe $F_{2|1} > 99.90\%$ for $d \geq 4w$. To calibrate the intensity crosstalk for these measurements, we per- form another experiment with $d = 0$ with attenuated intensity. $F_{1|2}$ measured with attenuated light confirms that our intensity crosstalk is in the regime of $\lesssim 8 \times 10^{-5}$ (Fig. 3c). This estimation of the intensity of crosstalk is also corroborated by our atomic physics simulations (Supplementary Note 5).

### Site-selective state measurement
State measurement of the ion qubits is achieved by detection of the state-dependent fluorescence[29]. The ions are excited by the probe light resonant to the $D_1^{(10)}$ cycling transition and the light they scatter is detected with limited efficiency by a measurement apparatus. While a longer probing duration ($\tau_d$) improves the measurement fidelity of the process qubit (until other effects such as off-resonant scattering[30] dominates), it also increases $P_{\text{AQM}}$ of the asset qubit for a given intensity crosstalk. Hence, in-situ mid-circuit measurement requires both a short $\tau_d$ and low $I_X$. For the fastest experimentally demonstrated measurement

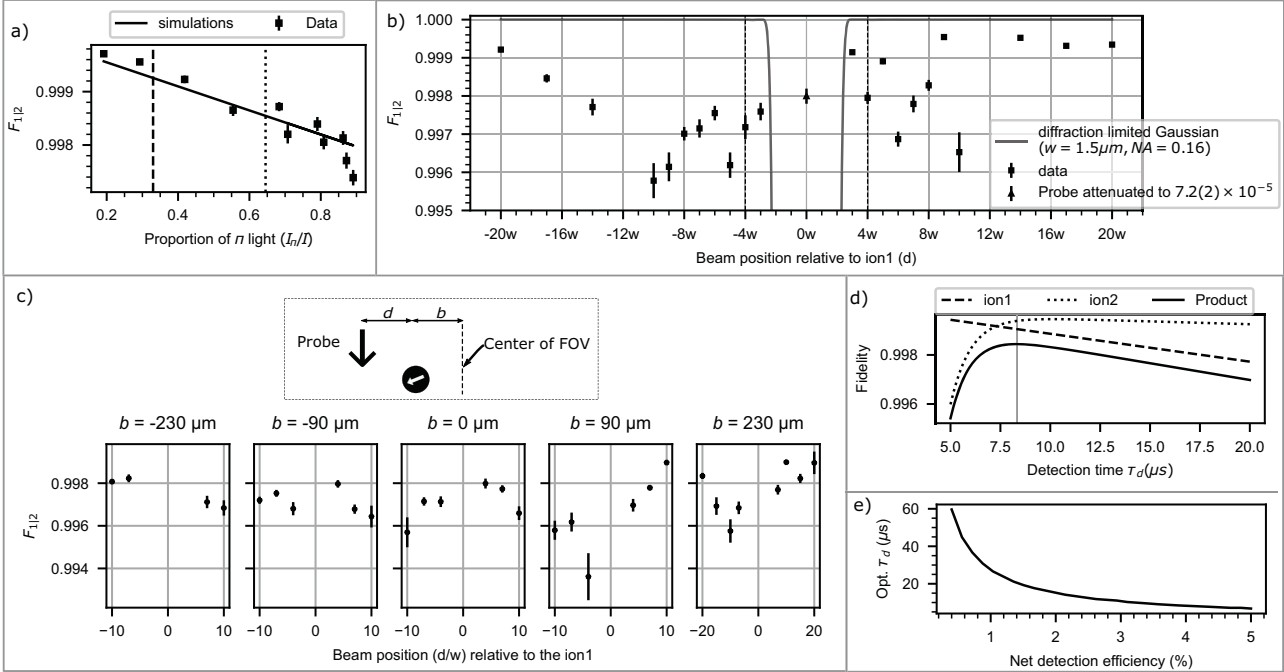

**Fig. 4 | Fidelity ($F_{1|2}$) of preserving ion1 for detection light at ion2 location. a** $F_{1|2}$ vs polarization of the detection probe light, showing that it is maximized for probe light with no $\pi-$ polarization. The dashed line represents the optimal polarization[31] for the process qubit (ion2) state-detection. The dotted line represents the polarization used to measure $F_{1|2}$ in figures **b**, **c**. Measured values of $F_{1|2}$ at $d = 4w$, shown in figures **a–c**, are for detection probe light of intensity $I = 1.25(16)I_{sat}$ applied for $\tau_d =$ 11 μs. Error bars in figures **a–c** denote standard deviation in estimating $F_{1|2}$, using 20 bootstrapping repetitions from 200 measurements (Supplementary Note 7). The estimated infidelity (1-$F_{1|2}$) due to the inherent decoherence of the qubit in the absence of the probe light is $< 3 \times 10^{-5}$. **b** $F_{1|2}$ vs the distance $d$ (case-B in Fig. 2b). For comparing the crosstalk $I_X$, we measure $F_{1|2}$ for a probe beam with relative intensity attenuated to $7.2(2) \times 10^{-5}$ addressing ion1 (triangle marker at $d = 0$). For comparison, $F_{1|2}$ is calculated(solid gray line) for a diffraction-limited (Numerical Aperture NA = 0.16) Gaussian beam of beam waist $w$. $F_{1|2}$ fidelity is $> 99.6\%$ for $d \geq 4w$. **c** Measured $F_{1|2}$ for various shifted locations of the ion from the center of the field of view (FOV). Here the center of FOV denotes the location at which the aberrations

have been characterized and compensated (see Methods). $F_{1|2}$ is preserved for a large FOV of 460 μm. **d** Calculated process qubit (ion2) detection fidelity[13,30] (Supplementary Note 6) and asset qubit(ion1) preservation fidelity ($F_{1|2}$) as the function of $\tau_d$ (detection time). Here, for estimating the process qubit (ion2) detection fidelity, we assume that the process qubit is illuminated with a detection beam of $I_2 = I_{sat}$ with optimal polarization and a measurement apparatus of net detection efficiency of 4%, compatible with the state-of-the-art experiments. We employ a photon count thresholding method to differentiate between $| \uparrow \rangle$ and $| \downarrow \rangle$ states. Furthermore, we use an algorithm that completes the detection process upon measuring the first photon, reducing detection time by a factor of 2[13,41,42]. For estimating $F_{1|2}$, we assume intensity crosstalk of $I_X = 5 \times 10^{-5}$, $I_2 = I_{sat}$, and optimal polarization for the process qubit state-detection. The vertical line at $\tau_d \approx 8.5$ μs represents the optimal detection time that maximizes the product of these two fidelities. **e** Optimal detection time (opt. $\tau_d$) as a function of the net detection efficiency of the measurement apparatus.

time of $\tau_d = 11$ μs[13], in-situ mid-circuit measurement with $P_{AQM} \sim 10^{-3}$ for the asset qubit is feasible, for our measured intensity crosstalk below $10^{-4}$ level. Note that the photon detection method employed in ref. 13 involves an efficient and low-noise detector, which is suitable for adaptation in current trapped ion experiments. Here, we report $F_{1|2}$ when a detection beam (with similar intensity, frequency, and polarization as in ref. 13) was applied on ion2 location for a time of 11 μs (Fig. 4).

$\sigma^+$ and $\sigma^-$-polarized $D_1^{(10)}$ light don't cause AQMs to the asset qubit in $S_{1/2}|F=1, m_F=0\rangle$ state, ignoring low probability off-resonant excitation (Fig. 3a). Hence, the asset qubit's fidelity is maximized with the least proportion of $\pi$ light (Fig. 4a). But the optimal polarization[31] for the highest scattering rate and hence the highest detection fidelity for ion2 is $I_\pi = I_{(\sigma^+)} = I_{(\sigma^-)}$.

We examine $F_{1|2}$ as a function of beam position relative to the ion ($d$) (Fig. 4b) using a detection beam in a case-B configuration. We find that the long coherence times ($T_2^*$) result in fidelities $F_{1|2} > 99.5\%$ for $d \geq 4w$ and $F_{1|2} > 99.9\%$ for $d \geq 20w$. Note that the polarization of the probe beam for these measurements is $I_\pi \approx 0.6$ and even higher fidelities could be achieved for the optimal detection polarization (Fig. 4a). This high fidelity $F_{1|2}$ is maintained in the measurements with ion shifted 100 μm and 200 μm away from the center of the field of view (FOV) (Fig. 4c), demonstrating that in-situ measurements are possible in a long chain of ions. Note that, for all the aforementioned

measurements, the aberration was compensated using the phase profile measured at $b$=0 (see Methods). Moreover, by compensating the aberrations using a phase profile measured at a different point located away from the center of the FOV, it is possible to achieve even higher fidelities at that specific point.

The detection-fidelity of the process qubit for a given detection efficiency (from a limited NA and photon collection loss) increases with increasing the detection time ($\tau_d$) (ignoring the off-resonant effects) whereas $F_{1|2}$ decreases (Fig. 4d). The optimal detection time depends on the relative importance of these fidelities in a given quantum algorithm. For example, one metric to find optimal detection time could be to maximize the product of these fidelities. This optimal detection time is highly dependent on the net efficiency of the detection apparatus (Fig. 4e).

## Discussion
In summary, we have demonstrated high fidelity in preserving an ion qubit while the neighboring qubit is reset or measured at a few microns distance. Our results are comparable to the state-of-the-art QIP experiments[12,13,32] that employ shuttling of qubits to be preserved away from reset or measurement laser beams by hundreds of microns distance.

Further, our protocol could be combined with other error-mitigation methods, such as shorter-distance shuttling or usage of a

different isotope of the same ion species, paving ways to reduce crosstalk errors compatible with quantum error correction protocols. Short-distance (tens of microns) shuttling would also improve the speed of the quantum algorithms and reduce errors from motional heating when compared to hundreds of microns shuttling used in current experiments. For a typical isotope shift of a few hundred MHz and our demonstrated $\lesssim 8 \times 10^{-5}$ intensity crosstalk, the $P_{AQM}$ for state reset and measurement can be reduced to the $10^{-6}$ level. Using a different isotope of the same ion species will also remove challenges, such as reduced motional coupling between ions of disparate masses[27,33,34] during mid-circuit sympathetic cooling and quantum gate operations.

Our crosstalk measurement scheme employs temporal separation of probe light illumination and detection of an ion qubit and hence overcomes sensitivity limitations due to unwanted background scattering of resonant light from optics leaking onto photon detectors in previous experiments[26]. This in turn allows measurement of crosstalk over a large dynamic range.

Ions are localized to < 100 nm at typical laser-cooling temperatures and trap frequencies, making it possible to characterize aberrations with the ion sensor for larger numerical aperture (NA) systems. With large NA, the beam waist $w$ decreases, thus the ion separations can be decreased without increasing $P_{AQM}$ to achieve higher qubit-qubit interaction strengths[35].

Our demonstrated high fidelity over a FOV of 450 μm corresponds to ∼ 50 ions in a linear chain for typical harmonic trapping parameters (radial trap frequency of approx. $2\pi \times 5$ MHz and axial trap frequency of approx. $2\pi \times 30$ kHz). The slight decay of fidelity away from the center of FOV can be compensated by recalibrating aberrations away from the center. However, even without extra calibrations, the fidelity $F_{1|2}$ can be maintained over the entire chain, as inter-ion separation away from the center of an ion chain also increases in a harmonic trap (from $4w = 6$ μm at the center becoming $\approx 10w = 15$ μm near the edge for parameters above[36]).

For typical radiofrequency ion traps (e.g., surface traps[37], 'blade' electrode traps[38]), NA > 0.5 is accessible for photon collection simultaneously with NA ∼ 0.3 (in a perpendicular direction) for optical addressing, allowing for independent optimization for photon collection and addressing. While high quantum efficiency detectors and negligible dark counts make ∼ 10 μs detection time possible[13], less-expensive photomultiplier tubes (PMT) can also allow ∼ 20 μs detection time (Fig. 4e)[13,30] under otherwise identical conditions for maintaining high asset qubit preservation fidelities of > 99.2%.

While the asset qubit coherence in our measurements is limited by intensity crosstalk, the $P^*_{AQM}$ from inter-ion scattering for state detection may be suppressed even further with the proper choice of the local magnetic field. For $^{171}Yb^+$, it is possible to suppress (see Methods) the intensity of $\pi$ light scattered from the process qubit in the direction of the asset qubit by aligning the magnetic field along the ion chain[39], thereby maximizing $F_{1|2}$ (Fig. 4a). In contrast, the optimal orientation of the magnetic field for state reset is perpendicular to the ion chain.

Comparing with the inter-ion scattering calculations in ref. 18, we find that the insensitivity to $\sigma^{\pm}$ photons (for state-detection through $D_1^{(10)}$ transition) for $^{171}Yb^+$ gives about ∼2 times reduction in $P_{AQM}$ compared to some other species, such as $^{40}Ca^+$ that is affected by all polarizations. Our scheme of obtaining low $P_{AQM}$ can be easily adapted to other ion species or different QIP platforms that benefit from high-quality individual optical addressing.

## Methods

### $P^*_{AQM}$ due to inter-ion scattering

Consider two ions (ion1 and ion2) separated by a distance $a$ in an ion chain. An ideal resonant laser beam illuminates ion2 without leaking any photons onto ion1. Ion2 scatters photons at a rate $\Gamma_{sc}(ion2)$, a portion of which are incident on ion1. The effective intensity of light on ion1 from these scattered photons is denoted by $I_{ab}(ion1)$. The relation between $\Gamma_{sc}(ion2)$ and $I_{ab}(ion1)$ is given as

$$I_{ab}(ion1) = f_{pol} \, f_{angle} \, \frac{h\nu \Gamma_{sc}(ion2)}{4\pi a^2} \tag{2}$$

Here, $\nu$ represents the frequency of the scattered light, $f_{pol}$ denotes the fraction of light whose polarization affects ion1, and $f_{angle}$ represents the angular dependence of the scattered light.

When ion2 is illuminated with an ideal state-detection beam, it emits light of all polarizations. However, only the $\pi$ polarization causes $P^*_{AQM}$ in ion1. Therefore, we have $f_{pol} = 1/3$. The angular dependence of light scattered in $\pi$ polarization by ion2 in the direction of ion1 is given by $f_{angle} = \sin^2(\theta)$[39], where $\theta$ is the angle between the magnetic field and the ion chain. In our setup, since the magnetic field is perpendicular to the ion chain, $f_{angle} = 1$. However, by choosing the magnetic field along the ion chain, $f_{angle}$ can be suppressed to zero. For a state-detection probe beam with an intensity of $I_2 = I_{sat}$, we use the optimal scattering rate of ion2[40] to estimate $\Gamma_{sc}(ion2)$. Assuming an inter ion spacing of 6 μm, we estimate that $I_{ab}(ion1) \approx 9.5 \times 10^{-6} I_{sat}$. This results in a $P^*_{AQM} = 2 \times 10^{-4}$ for 11 μs state-detection. If the intensity crosstalk can be maintained at the $1 \times 10^{-4}$, the inter-ion scattering will be the limiting factor below $a = 1.3$ μm. Note that this will require a probe beam with even smaller beam waste and hence larger numerical aperture ( ∼ 0.7).

For a state reset operation on ion2, $f_{pol} = 2/3$ since light with both $\sigma^+$ and $\sigma^-$ polarizations affect ion1. Additionally, for the case of a magnetic field perpendicular to the ion chain, we have $f_{angle} = 1/2$[39]. We estimate that the average $I_{ab}(ion1) \approx 1.3 \times 10^{-6} I_{sat}$ for the state reset operation. This results in a $P^*_{AQM} = 1 \times 10^{-5}$ for state reset.

### Aberration correction

We characterize optical aberrations in the entire beam path in terms of a Fourier plane (FP) phase map. An amplitude hologram on a Digital Micromirror Device (DMD) in the FP allows us to control the amplitude and phase of the diffracted light. The relative optical phase between two FP 'patches' is measured from the interference of beams that are diffracted from these patches. We use a single ion as a quantum sensor to measure this interference signal. We use an optical pumping beam on the ion, initialized in state $|\uparrow\rangle$, and observe state-dependent fluorescence signal from the ion as it gets pumped into state $|\downarrow\rangle$. By varying the phase of one of the FP patches and observing ion fluorescence for a fixed optical pumping time, we extract the interference profile and hence the relative phase. This approach is highly sensitive, as only a few photons are needed for optical pumping, allowing us to map out the phase diagram for the entire FP with very low optical power ( ∼ 200 μW of 369 nm light). By decoupling the probing and measurement, we achieve a higher signal-to-noise ratio compared with our previous approach[26], where an unwanted scattering of the probe beam from optics leaking onto the detector was a limiting factor. The aberrations are then compensated by generating the corrective hologram on the DMD using an iterative Fourier transform algorithm (IFTA)[26].

### Intensity and polarization calibration

We collect the ion fluorescence time-series data from many optical pumping experiments, where we controllably vary the relative optical power and polarization of the optical pumping light between experiments. Trends in these time-series data are fitted by numerical simulation to extract the saturation parameter and polarization of the light illuminating the ion (Supplementary Note 4D).

## Data availability

The data sets generated and/or analyzed for Figs. 1–4 during the current study, along with the plotting code, are provided in the Source Data file. Source data are provided with this paper.

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

## Acknowledgements

We thank Yu-Ting Chen, Crystal Noel and Crystal Senko for scientific discussions. We acknowledge financial support from the Canada First Research Excellence Fund (CFREF), the Natural Sciences and Engineering Research Council of Canada (NSERC) Discovery program (RGPIN-2018-05250), the Government of Canada's New Frontiers in Research Fund (NFRF), Ontario Early Researcher Award, University of Waterloo, and Innovation, Science and Economic Development Canada (ISED).

## Author contributions

S.M, C-Y.S., A.V., L.H., and J.Z. performed the experiments following an initial feasibility study by R.H., C-Y.S., and S.M. S.M., N.K., and D.M. performed theory calculations, numerical simulations, and analyses. S.M., A.V., and R.I. wrote the manuscript with inputs from all authors. All the authors contributed to the scientific discussions. RI supervised the whole project.

## Competing interests

The authors declare no competing interests.
