## [Peer Review File · Nature Communications]

Preserving a qubit during state-destroying operations on an adjacent qubit at a few micrometers distanceReviewers' comments:

Reviewer #1 (Remarks to the Author):

The experimental results from Motlakunta et al. address the topic of interaction with an atomic qubit, in this case a trapped ion, using a laser while at the same time avoiding incoherent interaction on a secondary "asset" qubit during two specific operations typical of quantum information processing, reset and detection/measurement.

The approach used is based on a DMD, which uses an engineered wavefront to produce a maximum of intensity in the laser light only at the position of the process ion. This approach differs from the more "common" approach of focusing the beam with a tight spot (such as done in some of the cited works) by correcting the wavefront of the laser to reach diffraction limited performance. The experimental setup and the technology used has already been used and demonstrated already in reference 28, by the same authors, to characterize single ion addressing crosstalk during coherent operations. The technical work in this aspect does not present any novelty and it is a follow up work.

On the technical side when selecting the time for the "detection" experiment the authors chose a time which is one of the shortest ever used for a trapped ion system. This time is far from the time actually used for their own detection protocol (1500 microseconds, supp. inf. B) or from what is also used in other experiments (hundreds of microseconds or more). This strongly limits the practical application in the specific case of measurement to only those fast systems. The reset process instead happens in the timescale of few microseconds normally in all trapped ion systems.

The authors take care of considering also inter-ion scattering error, which at the separations investigated (6 μm) can be neglected in respect to the errors induced by their addressing technique. The manuscript overall is full of information which sometime is of difficult access due to most of it being located in the supplementary information. I recognize that the length limit poses a deep challenge for such a technical manuscript and it is clear in some point that the necessary information has been moved.

Due to a lack of technical improvement from reference 28, I think that this work would benefit from being published in a longer format on a different journal.

minor formatting comment:

general lack of uniformity in the use of capitals for "(see methods)" seen in all its 4 possible iterations.

Comments by line:

Line 18: maybe add between parathesis the detection time of 11 us from reference 3. I was already wondering what the time was and it could put already things in context to have the information in the abstract.

Figure 1: I feel that the plots a and c should be exchanged in size given that a qualitative figure could be understood even if smaller.

line 44: definitions should feature in the main text if they are also used there.

line 68: I am bit baffled by the fact that the first cited reference is number 4 instead of number 1.

line 153: please specify to look in the supplementary information for more info on the experimental setup. The section is present but not referred to.

line 180: if possible can you provide a number for T^*_2 ? Tens of ms doesn't feel like a time scale where a measurement is difficult for trapped ions.

line 210: why $T \approx 0$ instead of equal to? If the incoherent beam is inactive it is possible to get the expected contrast by Rabi oscillations at the correct microwave detuning.

line 224: I suggest changing "parked on" to "addressing"

line 225: I suggest changing "parked" to "located"

line 231: Please once again point to the supplementary material. I was wondering if you were moving the beam or the ion which is not clear unless you read supp. inf. 5.

line 306: please add the definition of "I" in the text. Most people expect it to be the intensity but it should nevertheless be defined.

Figure 3: what is the y axis scale in figure b)? I imagine it is probability, but it is clearly missing a label and a unit

figure 4) caption, line 264: which parts are you referring to "a-c" ? given that the font is italics and not bold I imagine it is not figure 4a),4b) and 4c). if it is please write it in the same format i.e. "4a)-4c)" and exchange "parts" for "figures" to make it more clear.

line 355: typo achieved

line 363: missing period before "Moreover"

Supplementary material:

line 624: From my understanding this 10 kHz of detuning are an arbitrary choice to induce the state oscillations as a function of the probing/Ramsey time. Is that correct? If so maybe a sentence or two to explain it would be more helpful and leave the less attentive reader the reason for the frequency choice.

section D.5: Has the ion been displaced axially to maximize the fluorescence or has the beam been displaced? Seems like it is the second one from the text. If so, what is the minimum position resolution you can achieve? (i.e. minimum separation between two probe beam positions)

Reviewer #2 (Remarks to the Author):

This paper describes an experimental approach and demonstration that measures a trapped ion qubit using resonant fluorescence, while preserving the qubit stored in nearby ions. The experiments were enabled by creating a carefully engineered beam profile to focus the laser beam that drives the resonance fluorescence on the qubit to be read out, while keeping the inter-ion spacing at a few microns in the chain. The experimental demonstration indicates that selective detection of ion qubits in a chain is feasible without deciphering nearby qubits. The experimental demonstration is of interest in quantum computing research community, as the conventional wisdom is that selective qubit detection in an ion chain requires either extensive shuttling to isolate the qubits being detected, or different qubit states (or even species) had to be utilized. I believe this paper merits publication in Nature Communications, provided the following points are addressed by the authors.

1. The accidental quantum measurement (AQM) of neighboring qubits at very low levels will be dictated by the amount of finite intensity crosstalk (amount of detection beam impinging on the neighboring qubits). In this experiment, the beam is created using the DMD-based optical system. Experimental data shown in Figs 3b and 4b indicates that there is finite fidelity degradation of the qubit many beam waists away (up to 20 beam waists), deviating significantly from the diffraction-limited Gaussian beam predictions. This indicates that some fraction of the detection beam impinges on qubits in the chain at very low intensity levels (crosstalk levels reported here). Do the authors have

an explanation of the origins of this level of crosstalk? Is this coming from imperfect Gaussian beam formed by the DMD-based optical system, or by stray scattering of the detection beam in the system (vacuum chamber or from other nearby structures)? It is important for the authors to check the actual intensity profile of the detection beam created by the DMD-based optical system (in case this optical system poses a limit on the crosstalk), with at least 50dB of dynamic range. While Fig. S5 and S7 do not provide this level of dynamic range to characterize this level of spillover, their camera system shown in Fig. S6 seem to indicate that they should be able to confirm that.

2. In Line 402, they indicate the crosstalk measurement done here employs temporal separation of probe light that helps overcome the background scattering. (1) Does the unwanted background scattering affect the fidelity of the qubit state detection itself (dark state being considered bright)? What's a detection fidelity limit that can be imposed in their current experimental setup due to this effect? (2) Does the unwanted background scattering contribute to AQM in a detection situation (as per questions raised in previous paragraph)?

3. It looks like the "beam steering" (i.e., moving the detection beam wrt the ions) is performed by the DMD-based optical system. Given the DMD is a digital device and the beam steering is done in an analog fashion, I wonder if the profile of the beam, especially the low intensities at many beam waists away (presumably formed by undesired scatter or diffraction by the DMD devices) change as the beam is "steered" (e.g., as shown in Fig. S7).

4. There are some inconsistencies that should be addressed in their presentation:

a. The horizontal label of Fig. 4e says "Net quantum efficiency," while the caption says "net detection efficiency." Quantum efficiency tend to indicate the efficiency of a detector registering a photon (probability of signal out when one photon is incident), while the overall efficiency can include the fraction of the light collected by the optics, etc. The authors should clarify the definition both terms if they choose to use both, and explain the differences if indeed they are referring to quantum efficiency.

b. Same comment for the term "high quantum efficiency" comment on line 434.

c. In line 599 (Supplementary Information), the authors should use the same symbol for the beam waist (italicized w) as used in other places in the manuscript.

Reviewer #3 (Remarks to the Author):

The authors analyze the detrimental effect on qubit coherence caused by the mid-circuit detection and reset of a neighboring qubit. The main motivation behind this work is to study these unwanted effects in the case where the quantum register is composed of a string of identical ions (i.e. same species and isotopes). For this particular work, the qubit is encoded in two specific levels of the hyperfine ground-state of a ytterbium ion. They identify two main sources of decoherence. The first is inter-ion scattering, which relates to the absorption of a photon emitted by the neighboring being detected or reset. The analysis in this case is performed with two ions separated by approximately 9 μ m (about 6 times the waist of the addressing beam). The second source of decoherence is imperfect optical addressing and it induces qubit decoherence due to the unwanted interaction of a qubit with a detection beam. For this study the authors only work with a single ion and study the decoherence effect as a function of distance between the ion and the center of the laser beam.

Here below some major considerations about the paper:

1. Throughout the paper, the authors seem to hint (lines 95-96 and 121-124 and in the conclusion part) that their work will solve the issue of mid-circuit detection with ancilla qubits or the need for other techniques like shuttling or mixed-species operations, thus greatly reducing resources. The work

from the authors certainly doesn't solve the issue of ancilla qubits. Ancilla qubits for mid-circuit operations will be necessary. Think, for example, of quantum error correction. Without ancillas, there will be no way of measuring error syndromes without completely destroying the information stored in the data qubits. I also find odd what is said in line 90 where citation 6 is used as an example where ancillas of the same species are adding extra complexity to the system for readout. Provided what I wrote above that ancilla will be necessary for a large class of QIP experiments requiring mid-circuit operation (even in the case where one uses the technique put forth by the authors), I think that line 90 completely misuses the citation. In citation 6 extra ancilla qubits were used explicitly to avoid mid-circuit operation through the use of space-time duality (a clever trick introduced in various theoretical proposals especially targeting superconducting qubits realization of measurement-induced phased transitions)

2. While the study of imperfect optical addressing is performed in more detail, I find the analysis of the first decoherence source (i.e. inter-ion scattering) lacking depth. The study is done at a single distance (9um). Such ion-ion spacing is on the far side of the spectrum of the trapped-ion architectures I am aware of, where the ion-ion spacing is typically in the order of 5 um (especially for short ion chains with lighter ion species than ytterbium). While I understand that experimental measurements might be hard to make given the constraints of the setup I still think that it would be useful to have a theoretical analysis of this source. Questions that I would find interesting to address are: What is the expected infidelity as a function of distance (in the case where the second error source is completely neglected)? In other words, is this source something one should ever be worried about or not? Does it ever become relevant compared to unwanted illumination? The effect will also vary based on the quantization axis direction due to the anisotropic photon emission pattern for different polarization. How does the fidelity increase or decrease? I think these are important points that will be useful to take into account when designing new experiments.

3. The study of imperfect illumination is done in more detail and accounts for most of the author's results. In this part, I find particularly strange the presentation of the data. The datapoints have a very small error bar but there seems to be a large variability between neighboring datapoints. From Fig. S7 the beam doesn't seem to have a structure (though the data of the figure only extends to ~ 4 beam waists). What is the reason behind data variability? Are the error bars correctly calculated?

4. Based on equation 1 there should be a limit to the maximum achievable fidelity, set by the coherence time in the absence of any detection/reset light. Such value is never quoted in the main text and in the supplementary it is said to be much larger than 200 ms. What is it exactly? This is extremely important as it will say if the authors are simply measuring noise or if there is some unexplained structure to the data. Showing this maximum fidelity in the plots of Fig 3 and 4 will be very beneficial.

5. Related to the previous point I would expect that the maximum fidelity is around 99.8% based on data of Fig.4c as the beam is very far away from the ion (up to 230 um distance). So why in Fig.4b the authors measure even 99.99% with very small error bars? The figure caption says that when the beam was very far, the beam shape was not being corrected for aberrations. Is this enough to account for this discrepancy? It would require more supporting data, otherwise, it is hard to make any type of claim.

6. The authors seem to mix parameters based on their specific setup with parameters based on state-of-the-art experiments. This makes reading the paper quite confusing and makes some simulations hard to compare with the actual data they measured. Clearly, it is impossible in their setup to achieve high-fidelity detection in 10 us as the numerical aperture of their apparatus is only 0.12 as opposed to 0.6 (ie. the one used in the state-of-the-art setup quoted by the authors). How much longer is the detection time of the authors' setup? My hypothesis is at least 300 us. This will dramatically change the measured fidelities and will also affect the claim of realizing QIP experiments involving mid-circuit operations on long ion chains at the fidelity level quoted in the experiment.

7. Compared to the existing literature I have a hard time understanding which parts of the paper provided substantial elements of novelty.

Other than these points I found the text at times quite hard to follow. For example, there are various

typos, like missing articles that would help the reader when fixed. I would also suggest putting citations at the end of a sentence for improved readability whenever possible.

Response to the Reviewers

Sainath Motlakunta, Nikhil Kotibhaskar, Chung-You Shih, Anthony Vogliano,
Darian McLaren, Lewis Hahn, Jingwen Zhu, Roland Hablützel, Rajibul Islam,
Institute for Quantum Computing and Department of Physics and Astronomy,
University of Waterloo, Waterloo, Ontario N2L 3G1, Canada

We thank the reviewers for taking the time to review our manuscript and raise important points about our experimental results. We have addressed the reviewers' comments in this document and have made the necessary changes to the manuscript. We hope that our response and the revised manuscript adequately answer their concerns, and that they will consider our manuscript for publication in *Nature Communications*.

Response to Reviewer #1's comments:

- 1) Reviewer-1:** The experimental setup and the technology used has already been used and demonstrated already in reference 28, by the same authors, to characterize single ion addressing crosstalk during coherent operations. The technical work in this aspect does not present any novelty and it is a follow up work.

Response:

We are surprised by this assessment from the reviewer. Reference 28 (C.-Y. Shih et al., npj Quantum Inf 7, 57 (2021)), and the current manuscript differ greatly in their scientific goals, the techniques and methodology used, and the quantitative results.

Reference 28 was a technical paper introducing holographic beam shaping technology for trapped ion experiments. On the other hand, the novelty of the present work lies in being the first reported demonstration of the feasibility of in-situ 'mid-circuit' measurement and reset (MCMR) within a system of identical trapped ion qubits. Solving the in situ MCMR problem required atomic physics explorations and over two orders of magnitude improvements in the measured intensity crosstalk at the ion compared to Ref. 28. Further, the result that the inter-ion scattering is not a limiting factor for in situ MCMR, verified by both theory and experiments in the current manuscript, is itself of high significance. Our results in current manuscript go against the conventional wisdom of the ion trapping community, as articulated in the comments from the second reviewer: "The experimental demonstration is of interest in quantum computing research community, as the conventional wisdom is that selective qubit detection in an ion chain requires either extensive shuttling to isolate the qubits being detected, or different qubit states (or even species) had to be utilized."

Reference 28 introduced a novel way to generate a binary amplitude hologram that surpassed the performance of previous hologram generation protocols. With this protocol, we demonstrated $<10^{-4}$ relative intensity crosstalk measured on a camera. This measurement was performed in a system of very low NA (~ 0.02) at an 'intermediate image plane' before the vacuum system. Whereas the measured intensity crosstalk by a $^{174}\text{Yb}^+$ ion inside the vacuum system was limited to $\sim 10^{-2}$ (Fig. 3f and 4d of Ref. 28), not suitable for in situ MCMR (Fig. 1c of the current manuscript). Note that the 10^{-2} measured

crosstalk by the ion was despite aberration sensing by the ion itself and the aberration compensation by the hologram.

We would like to emphasize that obtaining diffraction-limited performance, as we did for the Doppler cooling beam at the ion in Ref. 28, guarantees confinement of most of the energy within the Airy Disc. However, it does not guarantee ultra-low crosstalk beyond the Airy Disc where the neighboring ion is located. Thus, from the results of Ref. 28, it was not obvious that in situ MCMR would be possible.

In fact, the techniques used in Ref. 28 were the limiting factor in achieving the necessary performance. Both the accuracy of the aberration phase map and the intensity crosstalk measurement were limited by the fluorescence measurement technique on $^{174}\text{Yb}^+$. We solve the crosstalk measurement problem in our current manuscript by utilizing the over five orders of magnitude separation in the timescales – from optical pumping to qubit coherence time - to control hyperfine qubits in $^{171}\text{Yb}^+$. The qubit acts as a high dynamic range sensor for intensity crosstalk measurement, surpassing previous limitations in Ref. 28. The use of optical pumping method (instead of fluorescence from Doppler cooling beam in Ref. 28) allowed more precise aberration sensing that is fast and highly sensitive to optical fields. The temporal separation of the aberration measurement from the fluorescence detection enhanced the sensitivity further. With the new techniques, the Fourier plane resolution in the aberration sensing increased by 16 times compared to Ref. 28. We verified using numerical simulations that this enhancement was crucial to achieve $<10^{-4}$ intensity crosstalk at the ion.

In summary, while Ref. 28 showcased an enabling technology, the feasibility of in situ MCMR was not self-evident in presence of inter-ion scattering, imperfect beam shaping, and other possible background scattering in the vacuum environment of the ion, until we demonstrated its feasibility in the current manuscript.

To emphasize these differences, we have added the above points in the main text as follows:

“We employ a holographic system for precise control of the laser beam's wavefront, correcting aberrations in the optical system. Our relative intensity crosstalk at the ion is a hundred times lower than what was attained in previous efforts using this technology. This reduction in crosstalk is crucial for high-fidelity, in-situ mid-circuit measurements and resets, and arises from two key advancements in our methods. Firstly, we utilize the qubit state's high sensitivity to a reset beam for intensity measurement, offering enhanced precision and spatial resolution for aberration characterization. Secondly, we assess intensity crosstalk using the wide dynamic range of the coherence time T_2^* , obtained from Ramsey interferometry on the qubit. The coherence time T_2^* varies greatly with beam intensity, turning the qubit into a highly efficient intensity sensor with a dynamic range exceeding 50 dB. This approach verifies the extremely low crosstalk.”

- 2) Reviewer-1:** On the technical side when selecting the time for the "detection" experiment the authors chose a time which is one of the shortest ever used for a trapped ion system. This time is far from the time actually used for their own detection protocol (1500 microseconds, supp. inf. B) or from what is also used in other experiments (hundreds of microseconds or more). This strongly limits the practical application in the specific case of measurement to only those fast

systems. The reset process instead happens in the timescale of few microseconds normally in all trapped ion systems.

Response: We agree with the reviewer that the processes of mid-circuit measurement and reset (MCMR) are difficult. Our approach requires fast detection time, which has been experimentally demonstrated (S. Crain et al., *Commun Phys* 2, 97 (2019)) using a commercially available detector. In that demonstration, the scattered photons were collected using a NA = 0.6 objective, which is also commercially available. Many existing traps, such as surface traps and 'blade' traps allow for such high NA photon collection in addition to sending a probe beam in the orthogonal direction. Our trap geometry (a four-rod trap) itself enables photon collection from two opposite directions, each with a numerical aperture of 0.4, perpendicular to the probe light. While our present hardware is not configured to utilize this feature, collecting photons from both 0.4 NA directions would allow for a collection efficiency like that of a system with an NA of 0.6.

We emphasize that, although upgrading the hardware of current traps like ours to achieve $\sim 10 \mu\text{s}$ detection time might be limited by economic or financial constraints, other methods for MCMR operation have even stricter needs. These include specialized traps capable of low-noise shuttling, that remains a subject of ongoing research and development. Working with multiple species (or isotopes) of ions for MCMR also requires additional trapping controls that are not available for many traps.

Further, in Fig. 4d of our manuscript, we provide the fidelities of both the asset and process qubits for total detection efficiencies of the imaging system. Quantum information processing algorithms can optimize the relative importance of the process qubit detection fidelity vs asset qubit preservation fidelity.

- 3) **Reviewer-1:** general lack of uniformity in the use of capitals for "(see methods)" seen in all its 4 possible iterations.

Response: We appreciate the reviewer for bringing this to our attention, and we have addressed the issue by making the necessary corrections to the manuscript.

- 4) **Reviewer-1:** Line 18: maybe add between parathesis the detection time of 11 us from reference 3. I was already wondering what the time was and it could put already things in context to have the information in the abstract.

Response: We appreciate the reviewer for bringing this to our attention. The necessary changes have been made in the abstract.

- 5) **Reviewer-1:** Figure 1: I feel that the plots a and c should be exchanged in size given that a qualitative figure could be understood even if smaller.

Response: We thank the reviewer for pointing this out, we have made the changes in Figure 1.

6) Reviewer-1: line 44: definitions should feature in the main text if they are also used there.

Response: We have incorporated the definition into the main text. Please refer to line 115 in the modified manuscript for the updated content.

7) Reviewer-1: line 68: I am bit baffled by the fact that the first cited reference is number 4 instead of number 1.

Response: We appreciate the reviewer for highlighting this concern. The issue has been addressed in the manuscript, and as a result, there have been modifications to the reference numbers and the line numbers.

8) Reviewer-1: line 153: please specify to look in the supplementary information for more info on the experimental setup. The section is present but not referred to.

Response: We appreciate the reviewer for bringing this to our attention. The issue has been addressed in the manuscript, and the corresponding correction can be found in line 163 of the updated manuscript.

9) Reviewer-1: line 180: if possible, can you provide a number for T_2^* ? Tens of ms doesn't feel like a time scale where a measurement is difficult for trapped ions.

Response: We have incorporated the T_2^* value (> 200 ms) into the main text at line 225, which was previously mentioned in the supplementary information. It is important to note that our Ramsey measurements, as depicted in Figure S3, do not show any decay until 200ms, indicating that our T_2^* is significantly longer than 200ms.

10) Reviewer-1: line 210: why $T \approx 0$ instead of equal to? If the incoherent beam is inactive it is possible to get the expected contrast by Rabi oscillations at the correct microwave detuning.

Response: We appreciate the reviewer for making this comment. Since the local amplitude of the Ramsey fringe is assigned to the beginning of the time window, we have indeed used the Ramsey fringe contrast at $T=0$. We have accordingly made the amendment in the manuscript to $T=0$.

11) Reviewer-1: line 224: I suggest changing "parked on" to "addressing"

Response: We appreciate the reviewer for providing this suggestion, and we have implemented the change in the manuscript.

12) Reviewer-1: line 225: I suggest changing "parked" to "located"

Response: We appreciate the reviewer for providing this suggestion, and we have implemented the change in the manuscript.

13) Reviewer-1: line 231: Please once again point to the supplementary material. I was wondering if you were moving the beam or the ion which is not clear unless you read supp. inf. 5.

Response: We appreciate the reviewer for providing this suggestion, and we have implemented the change in the manuscript.

14) Reviewer-1: line 306: please add the definition of "I" in the text. Most people expect it to be the intensity but it should nevertheless be defined.

Response: We appreciate the reviewer for providing this suggestion, and we have incorporated the definition into the manuscript at line 319.

15) Reviewer-1: Figure 3: what is the y axis scale in figure b)? I imagine it is probability, but it is clearly missing a label and a unit

Response: In Figure 3b), the plot on the right shares the same y-axis with the plot on the left. We appreciate the reviewer for pointing this out, and we have clarified this by making changes to the caption of Figure 3.

16) Reviewer-1: figure 4) caption, line 264: which parts are you referring to "a-c" ? given that the font is italics and not bold I imagine it is not figure 4a),4b) and 4c). if it is please write it in the same format i.e. "4a)-4c)" and exchange "parts" for "figures" to make it more clear.

Response: We appreciate the reviewer for bringing this to our attention, and we have resolved the issue in the manuscript.

17) Reviewer-1: line 355: typo achieved

Response: We appreciate the reviewer for bringing this to our attention. We have addressed the issue in the manuscript.

18) Reviewer-1: line 363: missing period before "Moreover"

Response: We appreciate the reviewer for bringing this to our attention. We have addressed the issue in the manuscript.

Supplementary material:

19) Reviewer-1: line 624: From my understanding this 10 kHz of detuning are an arbitrary choice to induce the state oscillations as a function of the probing/Ramsey time. Is that correct? If so maybe a sentence or two to explain it would be more helpful and leave the less attentive reader the reason for the frequency choice.

Response: We appreciate the reviewer for bringing this to our attention. We have addressed the issue in the manuscript as follows: " Here, Δ_{uw} is chosen such that the time periods of the Ramsey oscillations are much smaller than the characteristic decay times (T_2^*), while ensuring that Δ_{uw} is smaller than the microwave-induced Rabi oscillation frequency of 100 kHz."

20) Reviewer-1: section D.5: Has the ion been displaced axially to maximize the fluorescence or has the beam been displaced? Seems like it is the second one from the text. If so, what is the minimum position resolution you can achieve? (i.e. minimum separation between two probe beam positions)

Response: Yes, as the reviewer pointed out, the probe beam was displaced, and the minimum position resolution depends on the Fourier plane resolution of the DMD, diffraction limit, and the beam pointing stability. We estimate that our position resolution is currently limited by the beam pointing stability, approximately 200 nm over a period of a several minutes, which can be further improved with better mechanical stability. Numerical simulations have confirmed that our IFTA protocol reliably achieves beam displacements that are 10 times smaller.

Response to Reviewer #2's comments:

1) Reviewer-2: 1. The accidental quantum measurement (AQM) of neighboring qubits at very low levels will be dictated by the amount of finite intensity crosstalk (amount of detection beam impinging on the neighboring qubits). In this experiment, the beam is created using the DMD-based optical system. Experimental data shown in Figs 3b and 4b indicates that there is finite fidelity degradation of the qubit many beam waists away (up to 20 beam waists), deviating significantly from the diffraction-limited Gaussian beam predictions. This indicates that some fraction of the detection beam impinges on qubits in the chain at very low intensity levels (crosstalk levels reported here). Do the authors have an explanation of the origins of this level of crosstalk? Is this coming from imperfect Gaussian beam formed by the DMD-based optical system, or by stray scattering of the detection beam in the system (vacuum chamber or from other nearby structures)? It is important for the authors to check the actual intensity profile of the detection beam created by the DMD-based optical system (in case this optical system poses a limit on the crosstalk), with at least 50dB of dynamic range. While Fig. S5 and S7 do not provide this level of dynamic range to characterize this level of spillover, their camera system shown in Fig. S6 seem to indicate that they should be able to confirm that.

Response: We appreciate Reviewer-2's thorough examination of our results and careful observations. We do not fully understand the origin of the light leakage many beam waists away, at these lower crosstalk levels. We conjecture that this could indeed be due to unwanted scattering in the vacuum chamber, from the trap electrodes, or the objective lens system itself. Small length scale surface imperfections in the Fourier plane (FP), such as variations in individual micromirrors, may also contribute to the long spatial wavelength features in the image plane. We indeed observed that the crosstalk saturated to a small finite value with distance, even in the intermediate image plane, as measured on a camera (Fig. 4c in C.-Y. Shih et al., npj Quantum Inf 7, 57 (2021)). It will be interesting to investigate whether combining the FP DMD set up with an additional spatial switch in an intermediate image plane would get rid of the finite background. In the future, it will also be instructive to build a mock vacuum system (i.e., with the vacuum windows as identical as possible to the one used in the real system) to measure the degradation in the beam quality with the vacuum window.

However, we would like to emphasize that our method of measuring the intensity crosstalk, by evaluating the qubit coherence, functions as an optical sensor with a dynamic range of more than 50 dB. This approach provides a direct probe of the optical landscape. While an independent confirmation with a camera sensor will likely offer additional insights into the beam profile at a larger spatial scale, we believe that our ion measurements provide more accurate measurements, especially in the region of interest.

- 2) **Reviewer-2:** 2. In Line 402, they indicate the crosstalk measurement done here employs temporal separation of probe light that helps overcome the background scattering. (1) Does the unwanted background scattering affect the fidelity of the qubit state detection itself (dark state being considered bright)? What's a detection fidelity limit that can be imposed in their current experimental setup due to this effect? (2) Does the unwanted background scattering contribute to AQM in a detection situation (as per questions raised in previous paragraph)?

Response: In our experimental setup, the DMD-generated probe beam and the photon collection share the same path. While we use a separate global state detection beam that is perpendicular to the photon collection. Therefore, to measure the 'asset' qubit fidelity, we needed to temporally separate the DMD-generated probe beam from the measurement step. This separation is crucial to prevent background scattering from the probe beam from affecting the measurement apparatus.

- A) Indeed, the undesired background scattering from the probe beam entering the measurement apparatus impacts the detection of the 'process' qubit state itself. Given that our photon collection and the probe beam share the same path, the effect of background scattering is significant. It's important to note that we don't explicitly perform measurements on the 'process qubit' due to this limitation. However, this constraint can be easily addressed by configuring a setup where the probe beam path and the photon collection paths are perpendicular to each other. This adjustment qualitatively preserves the feasibility of our results in safeguarding the asset qubit.
- B) Yes, as mentioned in the previous comment, unwanted background scattering could contribute to AQM of the asset qubits if the scattered light is sampled by the asset qubit. This issue can be addressed by selecting optics with low reflectivity at the probe beam wavelengths, thereby improving the overall accuracy of asset qubit preservation.

3) Reviewer-2: 3. It looks like the “beam steering” (i.e., moving the detection beam wrt the ions) is performed by the DMD-based optical system. Given the DMD is a digital device, and the beam steering is done in an analog fashion, I wonder if the profile of the beam, especially the low intensities at many beam waists away (presumably formed by undesired scatter or diffraction by the DMD devices) change as the beam is “steered” (e.g., as shown in Fig. S7).

Response: Small shifts of the beam (\sim the beam waist, w) can be accomplished by adding a ‘tilt’ term to the hologram. In our measurements on the camera at IP1, we do not observe measurable changes in the beam profile many beam waists away while shifting the beam. For larger shifts, we use the iterative Fourier Transform Algorithm (IFTA) to generate the beam. The IFTA maintains the target crosstalk within a spatial window of interest, which spans $\sim 100w$ along the ion chain. We do not observe an appreciable change in crosstalk with ions when the beam is steered. This can be seen in Fig. 3c, where, for example, at a distance greater than $16w$, the data points are comparable within the error bars.

4) Reviewer-2: 4. There are some inconsistencies that should be addressed in their presentation:

Reviewer-2: a. The horizontal label of Fig. 4e says “Net quantum efficiency,” while the caption says “net detection efficiency.” Quantum efficiency tend to indicate the efficiency of a detector registering a photon (probability of signal out when one photon is incident), while the overall efficiency can include the fraction of the light collected by the optics, etc. The authors should clarify the definition both terms if they choose to use both, and explain the differences if indeed they are referring to quantum efficiency.

Response: We thank the reviewer for bringing the inconsistency to our attention. We have modified the figure to represent the net detection efficiency, which encompasses both detector efficiency and the fraction of light collected by the optics.

5) Reviewer-2: b. Same comment for the term “high quantum efficiency” comment on line 434:

Response: We thank the reviewer for pointing this issue, we have changed it to "high quantum efficiency detectors"

6) Reviewer-2: c. In line 599 (Supplementary Information), the authors should use the same symbol for the beam waist (italicized w) as used in other places in the manuscript.

Response: We appreciate the reviewer for bringing this to our attention. We have implemented the required changes accordingly.

Response to Reviewer #3's comments:

- 1) Reviewer-3:** Throughout the paper, the authors seem to hint (lines 95-96 and 121-124 and in the conclusion part) that their work will solve the issue of mid-circuit detection with ancilla qubits or the need for other techniques like shuttling or mixed-species operations, thus greatly reducing resources. The work from the authors certainly doesn't solve the issue of ancilla qubits. Ancilla qubits for mid-circuit operations will be necessary. Think, for example, of quantum error correction. Without ancillas, there will be no way of measuring error syndromes without completely destroying the information stored in the data qubits. I also find odd what is said in line 90 where citation 6 is used as an example where ancillas of the same species are adding extra complexity to the system for readout. Provided what I wrote above that ancilla will be necessary for a large class of QIP experiments requiring mid-circuit operation (even in the case where one uses the technique put forth by the authors), I think that line 90 completely misuses the citation. In citation 6 extra ancilla qubits were used explicitly to avoid mid-circuit operation through the use of space-time duality (a clever trick introduced in various theoretical proposals especially targeting superconducting qubits realization of measurement-induced phased transitions)

Response: We are sorry for the confusion. Our solution makes no claim about the requirement of ancilla qubits needed for error correction protocols. Instead, our in-situ MCMR approach eliminates the need for 'additional' ancilla qubits used for 'delayed measurements' as an alternative to mid-circuit measurements, as illustrated in C. Noel et al., Nat. Phys. 18, 760–764 (2022). We believe that the confusion may arise from our use of the terminology 'ancilla'. In the context of quantum error correction, the ancilla qubit used for syndrome measurements is equivalent to the 'process' qubit in our description.

We recognize that Noel et al. employ a clever technique involving delayed measurements to avoid mid-circuit measurements. However, this technique comes at an experimental cost – they need to use an additional ancilla qubit for every mid-circuit measurement. Quoting from Noel et al., "After each entangling gate, we add a measurement with probability P (Methods). Although mid-circuit readout of ion qubits is possible, we use ancilla qubits to defer readout. When a circuit calls for measurement, we entangle that qubit with an ancilla in a chosen measurement basis." This method wastes additional qubits, which is a valuable experimental resource. We also emphasize that these extra ions may introduce additional errors, as they introduce additional phonon modes that must be controlled for high fidelity quantum logic gates. Thus, from an experimental resource point of view, maximizing the number of usable and connected qubits is essential to scale quantum processors.

Conversely, in our approach of in-situ mid-circuit measurements, we do not need additional ancilla qubits for a given process qubit to be measured. With the capabilities of mid-circuit measurements and real-time classical feedback, the utilization of measurement ancilla qubits can be circumvented.

We agree with the reviewer that our use of the word ‘ancilla’ may confuse readers and hence have replaced it with ‘additional qubits’ in the manuscript.

- 2) Reviewer-3:** 2. While the study of imperfect optical addressing is performed in more detail, I find the analysis of the first decoherence source (i.e. inter-ion scattering) lacking depth. The study is done at a single distance (9 μ m). Such ion-ion spacing is on the far side of the spectrum of the trapped-ion architectures I am aware of, where the ion-ion spacing is typically in the order of 5 μ m (especially for short ion chains with lighter ion species than ytterbium). While I understand that experimental measurements might be hard to make given the constraints of the setup I still think that it would be useful to have a theoretical analysis of this source. Questions that I would find interesting to address are: What is the expected infidelity as a function of distance (in the case where the second error source is completely neglected)? In other words, is this source something one should ever be worried about or not? Does it ever become relevant compared to unwanted illumination? The effect will also vary based on the quantization axis direction due to the anisotropic photon emission pattern for different polarization. How does the fidelity increase or decrease? I think these are important points that will be useful to take into account when designing new experiments.

Response: We acknowledge the reviewer's observation regarding the experimental measurements for inter-ion scattering, which were primarily constrained by the limitations of our experimental apparatus, resulting in data obtained at a single inter-ion distance. However, we wish to direct the reviewer's attention to the Methods section in our manuscript, where we already provided a more detailed theoretical exploration of P_{AQM}^* resulting from inter-ion scattering.

We have illustrated that the intensity associated with inter-ion scattering, and consequently the infidelity (P_{AQM}^*), adheres to a scaling pattern inversely proportional to the square of the distance between the ions, when disregarding the intensity crosstalk from imperfect illumination. This assumes that the inter-ion separation is much larger than the wavelength (369 nm) of the scattered photons. Furthermore, we have investigated the dependence on the orientation of the quantization axis, considering anisotropic photon emission for various polarizations in both the detection and spin-reset processes.

Considering the angular dependence of the emission pattern of the process qubit, especially for the polarization (π) that influence the asset qubit in relation to the quantization axis of our experimental setup, we have calculated that the estimated value of P_{AQM}^* is roughly 2×10^{-4} . This is for detecting the state over a duration of 11 microseconds and with a spacing between ions of 6 micrometers. This value of P_{AQM}^* is 20 times smaller than the P_{AQM} of 4×10^{-3} achieved in our experiments at similar inter-ion distance, for an intensity crosstalk of 1×10^{-4} .

As $P_{AQM}^* \propto 1/a^2$, we estimate that $P_{AQM}^* = 4 \times 10^{-3}$ for $a = 1.3 \mu\text{m}$. Thus, if the intensity crosstalk can be maintained at the same value (i.e., 1×10^{-4}), the inter-ion scattering will be the limiting factor below $a = 1.3 \mu\text{m}$. Note that this will require a probe beam with even smaller beam waist and hence larger numerical aperture (≈ 0.7).

For state reset, the P_{AQM}^* is approximately 1×10^{-5} with an inter-ion spacing of 6 micrometers, which is 100 times smaller than P_{AQM} due to unwanted illumination. Thus P_{AQM}^* becomes limiting at much smaller inter-ion spacing well beyond the reach of current experimental trapped ion implementations.

To better illustrate the points mentioned above, we have modified the methods section.

- 3) Reviewer-3:** 3. The study of imperfect illumination is done in more detail and accounts for most of the author's results. In this part, I find particularly strange the presentation of the data. The datapoints have a very small error bar but there seems to be a large variability between neighboring datapoints. From Fig. S7 the beam doesn't seem to have a structure (though the data of the figure only extends to ~ 4 beam waists). What is the reason behind data variability? Are the error bars correctly calculated?

Response: We appreciate the reviewer's observation regarding the landscape of fidelities beyond a few beam waists. It is important to highlight that this observation underscores the sensitivity and remarkable dynamic range (> 50 dB) of the ion to sense the intensity using our Ramsey measurement scheme. In contrast, the measurements presented in Figure S7 are meant to estimate the probe beam waist and its location of the center. Therefore, this data was taken using a simpler population measurement (using 'optical pumping') instead of the Ramsey measurement, resulting in only ~ 10 dB of dynamic range. Thus, the plot in S7 does not show the features of the beam at low intensity levels.

Due to higher-order aberrations, the faint light intensity beyond a few beam waists fluctuates rapidly with distance. The low error bars result from the fact that, despite the beam profile changing rapidly with distance, it remains static and does not undergo further changes within the duration of a data set. Error bars denote standard deviations in estimating fidelities using 20 bootstrapping repetitions from 200 measurements. However, we do observe that the profile has slower drifts over multiple days, as can be seen from the variations in the profiles between Figures. 3 and 4, for which the data sets were taken five days apart without re-calibrating the aberrations. Our reported infidelities of 4×10^{-3} and 1×10^{-3} (for measurement and reset respectively) are conservative estimates, taking this drift into account.

- 4) Reviewer-3:** 4. Based on equation 1 there should be a limit to the maximum achievable fidelity, set by the coherence time in the absence of any detection/reset light. Such value is never quoted in the main text and in the supplementary it is said to be much larger than 200 ms. What is it exactly? This is extremely important as it will say if the authors are simply measuring noise or if there is some unexplained structure to the data. Showing this maximum fidelity in the plots of Fig 3 and 4 will be very beneficial.

Response: We appreciate the reviewer for bringing up this important point. As highlighted by the reviewer, we recognize the significance of characterizing the maximum achievable fidelity in the absence of any probe light. Figure S3 illustrates the Ramsey experiment measuring the coherence of the process qubit without any probe light. This measurement was conducted with Ramsey wait times extending up to 230 ms. As depicted in Figure S3, no observable decay was measured within this range. Extrapolating this plot, we estimate the inherent coherence time to be many seconds, which is consistent with the

long coherence time observed in other experiments for Yb+ qubits. We do not report the extrapolated coherence time in the manuscript, as this extrapolation has large uncertainties. Our measurements were limited to 230 ms due to practical considerations. Conducting experiments for longer durations is time-consuming, given that each point on the plot represents a repetition of 200 experiments.

The infidelity corresponding to a 230 ms coherence time (from Eqn 1) is $\approx 3 \times 10^{-5}$ (for a process time of about 11 μ s). Importantly, this value is orders of magnitude smaller than the infidelities shown in Figures 3 and 4, demonstrating that we are measuring infidelities larger than the inherent infidelity of the qubit.

To better illustrate the points above, we have made the following changes to the manuscript.

- 1) We added the following at the end of Appendix B of Supplementary Information.

“This large T_2^* corresponds to an infidelity $(1 - \text{adjfid}) < 3 \times 10^{-5}$.”

- 2) We have also added the estimate for infidelity in the captions of figures 3,4.

“The estimated infidelity due to the inherent decoherence of the qubit in the absence of the probe light is $< 3 \times 10^{-5}$ ”

- 5) **Reviewer-3:** 5. Related to the previous point I would expect that the maximum fidelity is around 99.8% based on data of Fig.4c as the beam is very far away from the ion (up to 230 μ m distance). So why in Fig.4b the authors measure even 99.99% with very small error bars? The figure caption says that when the beam was very far, the beam shape was not being corrected for aberrations. Is this enough to account for this discrepancy? It would require more supporting data, otherwise, it is hard to make any type of claim.

Response: There seems to be a misunderstanding of Figure 4c. In this figure, we assess the fidelity $F_{1|2}$ when the probe beam illuminates an ion placed far from the center of the field of view (FOV). Our intention is to demonstrate the efficacy of our protocol while addressing a large chain of ions. To achieve this, we relocate the ion from the center of the FOV to different positions denoted by the parameter b . Subsequently, we position the probe beam at a distance d away from this displaced ion, as illustrated in the inset of Figure 4c.

It is essential to clarify that the beam is not fixed 230 μ m away from the ion; rather, the ion is positioned 230 μ m (as indicated by $b=230 \mu$ m) away from the field of view (FOV) in the last panel. The beam is then steered around this ion. The observed decrease in fidelity when both the ion and the beam are distant from the center of the FOV can be attributed to the fact that aberrations were only calibrated at the center of the FOV.

Also, note that in Fig. 4b (corresponding to $b=0$), our reported fidelities at 20w are $\approx 99.9\%$, and not 99.99% as the reviewer commented.

- 6) **Reviewer-3:** 6. The authors seem to mix parameters based on their specific setup with parameters based on state-of-the-art experiments. This makes reading the paper quite confusing and makes some simulations hard to compare with the actual data they measured.

Clearly, it is impossible in their setup to achieve high-fidelity detection in 10 μs as the numerical aperture of their apparatus is only 0.12 as opposed to 0.6 (ie. the one used in the state-of-the-art setup quoted by the authors). How much longer is the detection time of the authors' setup? My hypothesis is at least 300 μs . This will dramatically change the measured fidelities and will also affect the claim of realizing QIP experiments involving mid-circuit operations on long ion chains at the fidelity level quoted in the experiment.

Response: We appreciate the reviewer's concerns regarding the mix of parameters in our setup and state-of-the-art experiments. In this manuscript, our focus is to demonstrate the preservation of the asset qubit, as fast measurement of a process qubit, although difficult, is a solved problem in the community.

Even with the demonstrated short measurement times of 11 μs , the community did not attempt to realize in-situ MCMR capabilities. The current manuscript demonstrates that preserving an asset qubit during in-situ MCMR is indeed feasible, if this fast measurement scheme is adopted. Other than minimizing the intensity crosstalk to $<1 \times 10^{-4}$ level, we also show that inter-ion scattering rate is not a limiting factor.

We thank the second reviewer of our manuscript to articulate this, "The experimental demonstration is of interest in quantum computing research community, as the conventional wisdom is that selective qubit detection in an ion chain requires either extensive shuttling to isolate the qubits being detected, or different qubit states (or even species) had to be utilized."

Fast measurements, around 10 μs , (S. Crain et al., *Commun Phys* 2, 97 (2019)) require the collection of photons through a high numerical aperture and the use of a photon detector that is both highly efficient and low in noise. Several existing ion traps, including blade traps and surface traps, possess adequate numerical apertures for effective light collection and the delivery of the probe beam from angles that are orthogonal. It is also important to note that the numerical aperture of 0.12 mentioned in the reviewer's comment and our manuscript pertains specifically to the probe beam. Our trap geometry itself enables photon collection from two opposite directions, each with a numerical aperture of 0.4, perpendicular to the probe light. While our present hardware is not configured to utilize this feature, collecting photons from both 0.4 NA directions allows for a collection efficiency like that of a system with an NA of 0.6.

S. Crain et al., also used a low noise detector (a Superconducting Nanowire Single Photon Detector), which is commercially available. Thus, replicating their solution, while adding our demonstrated beam engineering capabilities, is feasible for a wide range of ion traps including ours. However, we acknowledge that this will require extra resources (which is an economic constraint beyond the scope of the manuscript).

7) Reviewer-3: 7. Compared to the existing literature I have a hard time understanding which parts of the paper provided substantial elements of novelty.

Response: We summarize the novelties of the current manuscript.

1. This is the first demonstration of preservation of an asset ion qubit with high fidelity while a neighboring process qubit is reset in a direct way without wasting valuable experimental resources. The results presented do not require any special ion trapping architecture such as those required for low noise shuttling or mixed species, thus enabling the existing experiments to tackle novel problems in quantum simulations such as preparation of non-trivial initial states, algorithmic many body state preparation experiments, observation of dissipation driven quantum phases and phase transitions. Furthermore, in-situ state reset provides a method for performing sympathetic cooling on a sub-system of an ion chain, crucial for scalable and large-scale Quantum Information Processing (QIP) protocols.
2. This manuscript experimentally demonstrates that in-situ mid-circuit measurement is feasible. The protocol requires a fast measurement time, which is a solved problem in the community, and can be replicated with commercially available hardware.
3. The manuscript presents both experimental and theoretical results that show that inter-ion scattering does not prevent high fidelity in-situ MCMR.
4. The current manuscript demonstrates over two orders of magnitude improvements in reducing the measured intensity crosstalk at the location of an ion, compared to previous demonstrations using holographic beam shaping techniques.
5. The technique to use a single ion's quantum state as a sensor to measure ultra-low intensities with a dynamic range of over 50 dB is novel.
6. This manuscript demonstrates that the high preservation fidelities can be maintained for a long (>50) chain of ions, this can be further extended by performing site dependent aberration compensation.

REVIEWER COMMENTS

Reviewer #1 (Remarks to the Author):

I would like to thank the authors for their answer. It has now become to me much more clear the technical improvements done in respect to ref [28]. Indeed, limiting the intensity crosstalk outside of the Airy disk provides an additional challenge, necessary due to it being the location of the 'process' ions in long Coulomb crystals. The reworked manuscript addresses more the technical modifications that have been required for the presented work when compared to ref [28].

The use of tightly focused laser beams in quantum computing for minimizing crosstalk in logical operations is a known topic and implemented already by many groups that adopts architectures with a large number of ions in the same potential well. It is indeed correct from the authors to point out in their rebuttal that this is the first demonstration of measurement and reset in the same conditions. Although the detection time is not a commonly diffused one it has indeed been demonstrated experimentally. Therefore, I would recommend publication in Nature Communications.

Minor comments:

Line 356: Both in the answer to me, to the other Reviewers and in the manuscript you point out that the detector for the $11\mu\text{s}$ detection time is commercially available. I know various SNSPD are, I am unsure about the specific one used for ref [13] to which you refer. In ref [13] from Crain et al, is specified that ". E.E.W., M.D.S., V.B.V., and S.W.N contributed to the design and fabrication of the SNSPD devices used in these experiments". Is that correct or are you aware of different information? In case it is necessary please remove from the line "which is commercially available".

Lines 436 and 485: field of view can be written with the already defined abbreviation in the manuscript of FOV

figure 4d) I would suggest extending the timescale to at least $100\mu\text{s}$, or if you feel like the extended timescale would lose the necessary resolution for the details in the few μs scale, I would then suggest to add an extended figure to the supplementary material. My reasoning is that this could potentially be of interest to groups with high efficiency imaging system but that are not interested in 10^{-3} detection infidelity and are willing to withstand larger errors.

Reviewer #2 (Remarks to the Author):

I raised a few technical questions in the first review as a condition for publication, and I am not convinced that a couple of these questions were adequately addressed in the author's response or revised manuscript. It is important the questions are addressed before publication is warranted.

1. The first unaddressed question is the origin of the residual fidelity degradation many beam waists away (in both figures 3c and 4b), which is orders of magnitude larger than what's expected (given that the expected crosstalk is so small). This will look extremely obvious if instead of plotting fidelity, the authors plot infidelity ($1-\text{fidelity}$) in the vertical axis on a log scale. There is a very clear mechanism that leads to these crosstalk which limits the fidelity $F_{\{1|2\}}$ at a fraction of a percent level, which is detrimental to all applications that the authors see the value of this scheme to be useful for (for example, quantum error correction). It is important the authors provide an estimate on the origin of these "large" errors, so there is a path to improving the setup in the future by the authors or other groups. For example, if the origin of the errors is unexpected specular reflection from chamber windows, these issue can be designed away in other setups or in the future. But if the origin of the errors is coming from the edge-scattering or diffraction from the DMD device themselves, then it might render this approach unusable if the crosstalk needs to be suppressed by another order of magnitude or two. While I do not anticipate the authors to necessarily present measurement data

where the crosstalk is suppressed further, providing the fundamental origin of the source is critical in evaluating the viability of their approach.

2. The second unaddressed question is actually much more concerning. It is about the question of why the authors had to rely on "temporal separation" of the probe light to overcome scattering. I raised this question because of the concern that in their setup (shown in Fig. S1), the probe beam is incident on the same objective used to collect the photon for qubit state detection. In this scenario, one cannot avoid the probe light scattering off the lens surfaces that form the imaging optics themselves, and the sensitive detector (PMT) is blinded by this. This means one cannot reliably achieve the qubit state detection. From the authors' answers, it looks like (1) the detection of the "process qubit" is NOT actually performed anywhere in the experiment because the setup cannot reliably achieve the measurement, and (2) only the "hypothetical" effect of addressing crosstalk on the "asset qubit" was measured in this paper.

If this indeed is the case, then I don't think the data supports the title of the paper, that they have shown "preservation of a qubit during adjacent measurement at a few micron distance." I suggest two things: (1) modify their title so it is not as misleading, and (2) explicitly point out the limitation of their current setup (can't do high fidelity state detection) either in the section that starts at line 390 or in Appendix A of the Supplementary Material.

3. On a minor note, the caption of Figure 1b point to the two sub-figures as "Left" and "Right", but here they are presented as "Top" and "Bottom". This should be addressed in the final form of the figure and captions to be consistent.

Addressing the questions 1 and 2 in the manuscript is essential in recommending publication for Nature Communications.

Response to reviewer comments

We thank the reviewers for their positive assessment of our manuscript and relevant comments. Below, we address their remaining concerns and hope that they will find our response and the modified manuscript suitable for publication in *Nature Communications*.

Reviewer #1 : Line 356: Both in the answer to me, to the other Reviewers and in the manuscript you point out that the detector for the 11 μ s detection time is commercially available. I know various SNSPD are, I am unsure about the specific one used for ref [13] to which you refer. In ref [13] from Crain et al, is specified that ". E.E.W., M.D.S., V.B.V., and S.W.N contributed to the design and fabrication of the SNSPD devices used in these experiments". Is that correct or are you aware of different information? In case it is necessary please remove from the line "which is commercially available".

Response: We thank the reviewer for this keen observation. The reviewer is correct that the SNSPD used in main text Ref [13] was custom-fabricated. We have accordingly made the change in the manuscript and removed the word "commercial" in line 356.

From interactions with colleagues who regularly use SNSPDs, we learn that commercial vendors can optimize the response of an SNSPD to the desired wavelength, if an appropriate material is available for that wavelength.

Reviewer #1 : Lines 436 and 485: field of view can be written with the already defined abbreviation in the manuscript of FOV

Response: We thank the reviewer for this suggestion. We have revised the manuscript to use the abbreviation "FOV" for "field of view" as previously defined.

Reviewer #1 : figure 4d) I would suggest extending the timescale to at least 100 μ s, or if you feel like the extended timescale would lose the necessary resolution for the details in the few μ s scale, I would then suggest to add an extended figure to the supplementary material. My reasoning is that this could potentially be of interest to groups with high efficiency imaging system but that are not interested in 10^{-3} detection infidelity and are willing to withstand larger errors.

Response: We thank the reviewer for this valuable suggestion. In response, we have added the extended timescale graph to the supplementary Sec. IX.

Reviewer #2 (Remarks to the Author):

Reviewer #2: 1. The first unaddressed question is the origin of the residual fidelity degradation many beam waists away (in both figures 3c and 4b), which is orders of magnitude larger than what's expected (given that the expected crosstalk is so small). This will look extremely obvious if instead of plotting fidelity, the authors plot infidelity (1-fidelity) in the vertical axis on a log scale. There is a very clear mechanism that leads to these crosstalk which limits the fidelity $F_{\{1|2\}}$ at a fraction of a percent level, which is detrimental to all applications that the authors see the value of this scheme to be useful for (for example, quantum error correction). It is important the authors provide an estimate on the origin of these "large" errors, so there is a path to improving the setup in the future by the authors or other groups. For example, if the origin of the errors is unexpected specular reflection from chamber windows, these issue can be designed away in other setups or in the future. But if the origin of the errors is coming from the edge-scattering or diffraction from the DMD device themselves, then it might render this approach unusable if the crosstalk needs to be suppressed by another order of magnitude or two. While I do not anticipate the authors to necessarily present measurement data where the crosstalk is suppressed further, providing the fundamental origin of the source is critical in evaluating the viability of their approach.

Response: We thank the reviewer for pointing this out and apologize for not providing quantitative estimates in our previous response. To understand the source of these errors many beam waists away, we have conducted high dynamic range (HDR) measurements at the intermediate image plane using a camera and extracted the intensity crosstalk and corresponding infidelities. These measurements are presented in Supplementary Sec. X.

The results from the intermediate image plane demonstrate that the holographic beam shaping technique using a DMD itself can produce at least 55 dB contrast in intensity in the image plane. The residual intensity at larger separations (e.g., 20 beam waists away) is influenced by the binarization errors from the discrete DMD mirrors. We mitigate these errors using an Iterative Fourier Transform Algorithm (IFTA). If one can faithfully relay the intermediate image plane distribution to the ion, the residual intensity crosstalk at larger separations will correspond to $< 3 \times 10^{-5}$ infidelity on the asset qubit while the process

qubit is optically pumped, and $< 1 \times 10^{-4}$ asset qubit infidelity for state detection of the process qubit (detection time 11 μ s).

The performance at the ion plane, as presented in Figs. 3c and 4b of the manuscript, shows deterioration from the measured values at the intermediate image plane. We attribute this degradation to specular reflections from the chamber windows. Another possible source of error is uncompensated higher order aberrations. Both can be mitigated in other experiments, by using better glass and coatings, and by higher-resolution aberration characterization using an ion.

We note that these errors can be further reduced by using an additional DMD as a spatial switch in an intermediate image plane, similar to the setup described by Zhang et al. (Optica, vol. 11, no. 2, 2024, pp. 227, DOI: 10.1364/optica.512155), which should significantly reduce the crosstalk many beam waists away.

Reviewer #2: 2. The second unaddressed question is actually much more concerning. It is about the question of why the authors had to rely on “temporal separation” of the probe light to overcome scattering. I raised this question because of the concern that in their setup (shown in Fig. S1), the probe beam is incident on the same objective used to collect the photon for qubit state detection. In this scenario, one cannot avoid the probe light scattering off the lens surfaces that form the imaging optics themselves, and the sensitive detector (PMT) is blinded by this. This means one cannot reliably achieve the qubit state detection. From the authors’ answers, it looks like (1) the detection of the “process qubit” is NOT actually performed anywhere in the experiment because the setup cannot reliably achieve the measurement, and (2) only the “hypothetical” effect of addressing crosstalk on the “asset qubit” was measured in this paper.

If this indeed is the case, then I don’t think the data supports the title of the paper, that they have shown “preservation of a qubit during adjacent measurement at a few micron distance.” I suggest two things: (1) modify their title so it is not as misleading, and (2) explicitly point out the limitation of their current setup (can’t do high fidelity state detection) either in the section that starts at line 390 or in Appendix A of the Supplementary Material.

Response:

The reviewer correctly noted that we did not perform high-fidelity state detection. This is due to the numerical aperture constraints of our current ion-trap setup, which do not permit high-fidelity readout. Consequently, we are unable to perform fast

measurements in a perpendicular direction while concurrently sending the individual addressing beam.

We appreciate the reviewer's suggestions and have made modifications to the manuscript title to make it not misleading. The new title is

“Preserving a Qubit During State-Destroying Operations on an Adjacent Qubit at a Few Micrometers Distance.”

To explain the title, we have also made the following changes in the abstract, line 11:

“Protecting qubits from accidental measurements is essential for controlled quantum operations, especially during state-destroying measurements or resets on adjacent qubits, in protocols like quantum error correction.”

In addition, we have included in supplementary Sec. I (line 37) the limitations of our apparatus that prevent us from achieving high-fidelity state detection.

“The optical access to the ion in our apparatus, perpendicular to the probe beam direction (z), is limited to an NA of less than 0.1. Due to this limitation of our ion trap apparatus, we are unable to perform high-fidelity state detection of the ions while they are being probed using the state-detection beam.”

3. On a minor note, the caption of Figure 1b point to the two sub-figures as “Left” and “Right”, but here they are presented as “Top” and “Bottom”. This should be addressed in the final form of the figure and captions to be consistent.

Response: We thank the reviewer for this observation. We have made the necessary changes in the manuscript to address this issue. The caption of Figure 1b now correctly refers to the sub-figures as "Top" and "Bottom" to ensure consistency.

REVIEWERS' COMMENTS

Reviewer #2 (Remarks to the Author):

The authors have addressed the questions and suggestions raised in the previous round of reviews, and I believe the current manuscript is in a form that can be accepted for publication in Nature Communications.